# Novelty Detection with Autoencoders for System Health Monitoring in Industrial Environments

**Francesco Del Buono [1], Francesca Calabrese [2], Andrea Baraldi [1], Matteo Paganelli [1] and Francesco Guerra [1,*]**

[1] Department of Engineering "Enzo Ferrari", University of Modena and Reggio Emilia, 41125 Modena, Italy; francesco.delbuono@unimore.it (F.D.B.); andrea.baraldi96@unimore.it (A.B.); matteo.paganelli@unimore.it (M.P.)

[2] Department of Industrial Engineering (DIN), University of Bologna, 40136 Bologna, Italy; francesca.calabrese9@unibo.it

[*] Correspondence: francesco.guerra@unimore.it

**Abstract:** Predictive Maintenance (PdM) is the newest strategy for maintenance management in industrial contexts. It aims to predict the occurrence of a failure to minimize unexpected downtimes and maximize the useful life of components. In data-driven approaches, PdM makes use of Machine Learning (ML) algorithms to extract relevant features from signals, identify and classify possible faults (diagnostics), and predict the components' remaining useful life (prognostics). The major challenge lies in the high complexity of industrial plants, where both operational conditions change over time and a large number of unknown modes occur. A solution to this problem is offered by novelty detection, where a representation of the machinery normal operating state is learned and compared with online measurements to identify new operating conditions. In this paper, a systematic study of autoencoder-based methods for novelty detection is conducted. We introduce an architecture template, which includes a classification layer to detect and separate the operative conditions, and a localizer for identifying the most influencing signals. Four implementations, with different deep learning models, are described and used to evaluate the approach on data collected from a test rig. The evaluation shows the effectiveness of the architecture and that the autoencoders outperform the current baselines.

**Keywords:** novelty detection; anomaly detection; autoencoder; predictive maintenance; Industry 4.0

## 1. Introduction

Malfunctions in manufacturing plant equipment may cause unexpected production stops, which are associated with huge costs, including the loss of production and time, the loss of effort to the identification of the failure's cause and repair, the waste of those products produced right after bringing back the system before normal operations due to low quality, costs of repairs, and deterioration of equipment [1]. In other words, maintenance directly impacts productivity since the failure of a component may cause unplanned production downtimes, the duration of which may vary depending on the type of needed action and the availability of spare parts. Besides, an item not working correctly may compromise the output quality of a production system and the safety of the working environment.

The digitalization of current manufacturing industries offers a remarkable opportunity for system health management [2]. Machinery is equipped with sensors able to collect data at high frequencies; machinery in the same shop floor communicate and are connected to a central server, where the data of machinery installed in other plants can converge. In other words, Industry 4.0 technologies, like Industrial IoT, cloud computing, edge computing, and Big Data Analytics, enable the collection of a large amount of data from online condition monitoring systems. The knowledge of the exact assets' health condition in a given time instant has become an essential driver in maintenance management, since it provides the opportunity of setting efficient, just-in-time, and just-right maintenance strategies, resulting

in the maximization of the production profits and the minimization of all cost and losses, including asset ones [3]. Smart factories implementing evolute maintenance strategies show a 25–30% decrease in maintenance activities, a 35–45% breakdown, a 20–25% increase in production, and an investment return [4].

The massive amount of collected data contains valuable information and knowledge supporting condition-based maintenance and health monitoring [5]. In this context, the transformation of raw data into knowledge is usually referred to as Prognostics and Health Management (PHM), which uses Big Data Analytics and Machine Learning (ML) algorithms to perform fault diagnostics and prognostics. In real scenarios, an Industrial PHM system should [6]:

1. Include fault detection algorithms that can detect anomalies in the streaming data;
2. Include fault diagnosis algorithms that can classify the detected changes;
3. Consider incremental learning to deal with unlabeled datasets and any novel operating conditions of the machinery.

In the last few years, a large number of approaches (partially reviewed in the related work section) have been proposed to deal with fault detection and diagnosis via the experimentation of supervised techniques for anomaly detection and classification. Only recently have aspects related to the concept drift and the concept evolution that may occur in streaming data acquired a paramount importance in PHM. Indeed, there are two typical problems to address in real-world scenarios. The first is that of industrial secrecy, for which the real operating conditions in which the machine operates, i.e., the recipes, are not intended to be explicit in the data analysis. The second is that of the change in the environmental conditions in which the machinery operates. The machines are tested in laboratories with conditions different to those of production. The plants are dissolved in different areas with different environmental conditions. In all these cases, the problem to be faced is that of "novelty detection".

Novelty detection is the task of recognizing that test data differ in some respect from the data that are available during training [7]. The functionalities for a novelty detection system can be, in principle, provided by a classification model built on the datasets describing the operation conditions of the machinery under analysis. Nevertheless, its construction should consider that that the training dataset may not include all possible conditions that a component may experience during its life. The goal then is to recognize unknown behaviors and distinguish them from anomalies. In addition, they have to be complemented by mechanisms to automatically re-train the model as the novel behaviors are detected [8].

In this paper, we introduce a PHM system to perform novelty detection, based on autoencoders (AEs). They are neural architectures that, in the encoding phase, compress the input data into a compact vector representation and, in the decoding phase, reconstruct the original data starting from this intermediate representation [9,10]. In the context of novelty detection, these architectures can be used to identify new operating conditions by analyzing the reconstruction error, i.e., the possible error made by the decoder in reconstructing the input data. If the error exceeds a specified threshold, the processed input does not refer to any condition encountered in the training phase. Otherwise, the decoder would have correctly reproduced the input. In particular, this paper introduces an architecture template that extends the usual AE capability to:

1. Act as a classifier by identifying the condition in which the component is working, both known and unknown.
2. Deal with variable sampling windows, which represent a critical aspect in streaming applications. Since the streaming inference may be considered automatic labeling [11], a change in component or system behavior should be detected as soon as possible to collect correctly labeled datasets. To this purpose, the ideal situation is to apply the model to each arriving point. However, given the high sampling frequencies of sensors, the algorithm's execution time may be larger than the time between the two consecutive samples. Therefore, identifying the optimal sampling window means

    finding a trade-off between the necessity of having a fast response and the necessity of having the algorithm's response before the arrival of the next point.

3.    Detect the most anomalous/distinctive sensor(s) for each state of a multivariate time series. Detecting a change in the data streams is not sufficient for fault diagnosis, since it does not provide any information about the detected novel behavior's location or cause. Therefore, a further analysis aiming to find which variable mainly contributes to the change in the data streams is necessary.

    A systematic study of the AE-based architecture template is proposed, by evaluating four implementations, each one with a different DL model (Fully Connected, CNN, LTSM, and BiLSTM), against an emblematic dataset composed of nine multi-variate time series. The dataset was generated through a test rig built in the Department of Industrial Engineering of the University of Bologna. It describes four operating conditions, one of them including anomalies. The dataset can therefore reproduce many of the operating conditions that need to be addressed by novelty detection approaches. The code implementing the AEs and used for the experiments is available at https://github.com/softlab-unimore/AE4ND (accessed on 10 April 2022).

    This is not the first paper using AEs in the novelty detection field. In Refs. [12,13], an AE has been introduced for fault detection scenarios. Nevertheless, the paper differs from the former for the experimentation of four DL models and the ability to perform novelty detection and from the latter for the unsupervised nature of our proposal. Our study confirms the finding of the other approach, that AEs are an effective solution to the novelty detection problem.

    The rest of the paper is organized as follows. In Section 2, related works on novelty detection will be revised. In Section 3, the proposed architecture template for Novelty Detection is described. Section 4 introduces the experimental environment and Section 5 the evaluation. In Section 6, we point out some of the lessons learned. Finally, conclusions and future research directions will be highlighted in Section 7.

## 2. Related Works

    Unlike anomaly detection and outlier detection, novelty detection aims to find a set of points not explained from the diagnosis model, instead of one single point differing from the nominal or known data [8]. In addition, novelty detection is also included in multi-class systems, where two or more normal classes exist, and a condition is considered novel if it differs from all of the known classes [14]. In other words, the problem becomes recognizing novelties and, at the same time, classifying the known instances into two or more diverse classes.

    To this aim, traditional classification and clustering-based approaches are adopted for system Health Monitoring in industrial environments. In these cases, novelty detection approaches may be classified according to two criteria: the learning paradigm, which can be supervised or unsupervised, and the necessity of a training phase before the streaming application. Hence, three different approaches can be distinguished:

1.    Classification-based with training on one or more classes (e.g., COMPOSE [15,16]);
2.    Clustering-based with training on one or more classes (e.g., OLINDDA, MINAS [17–19]);
3.    Clustering-based that can be applied from scratch (e.g., ADP [20]).

    In addition, classification-based approaches may adopt an incremental or non-incremental learning paradigm, depending on whether the classification model is re-trained when a novel class is detected. On the contrary, all clustering-based approaches adopt an incremental learning paradigm.

    More recently, deep-learning models—such as autoencoders networks [12,13,21] and Generative Adversarial Networks (GAN) [22]—have been used in this field. GAN-based approaches have been proposed as anomaly detection systems, to learn a latent feature space of a generative network G so that the latent space well captures the normality underlying the given data. Some form of residual between the real instance and the generated instance is then defined as an anomaly score. Nevertheless, the training of GANs

can suffer from multiple problems, such as failure to converge and mode collapse, which leads to large difficulties in training GAN-based anomaly detection models. Moreover, the generator network can be misled and generate data instances out of the manifold of normal instances, especially when the true distribution of the given dataset is complex or the training data contain unexpected outliers. Autoencoders have been experimented in the field of the novelty detection. In particular, ref. [12] proposes an unsupervised AE LSTM architecture for the novelty detection task, intending to identify novel conditions in the machinery, but without focusing on the ability to distinguish the existing operating ones. The main paper innovation consists of the introduction of a supervised neural network that analyzes the incoming time series, classifying the type of fault in a supervised manner. Instead, ref. [13] introduces an AE architecture that combines the novelty detection part with the diagnosis (classification) part of the operating state. The innovation of the proposed architecture is the combination of the two metrics in a single optimization function. The approach requires labels on the data. As highlighted in the introduction, our approach follows this line of research by introducing a template architecture based on an autoencoder implemented with different deep learning models able to recognize novel operating conditions and classify the current one among the ones learned in a unsupervised manner. It relies on online learning: once a novel status is found, a retrain is needed. It differs from the current approaches in depth and breadth. We differ in depth since we propose a technique that is able to detect novel operating conditions, classify the current conditions and discover the signal that has most contributed to the inference in an unsupervised manner. We differ in breadth since we provide an extensive evaluation that includes many operating conditions (with anomalies) and compares the results with other systems.

## 3. An Architecture Template for Novelty Detection

Being able to recognize novel operating states where the machinery is working and clearly separate them from anomalies is one of the main problems affecting predictive maintenance in modern industry. The existing approaches, partially reviewed in the related work section, monitor the input signals (frequently in the form of multivariate time series) and behave like a binary classifier, showing when a signal is representing a known or novel status of the device.

Nevertheless, with only this information, it is not possible to evaluate complex scenarios such as the ones used in production systems. An architecture template for novelty detection should be able to do the following:

- Analyze big and heterogeneous data. Sensors on the machinery generate a large and complex amount of data that make ineffective manual inspections and analyses by experts. Supervised approaches need labeled data, the production of which requires a tremendous user effort. In many cases, actual machine operating states may vary, requiring additional labeling steps and model retraining. In other cases, the approaches should work with data collected at different sampling frequencies.
- Identification of the real conditions in which the component is working. A binary classifier recognizes the difference between a novel or known operating status. An effective tool should also classify the possible real conditions and infer the one where the system is actually working.
- Explain the novelty. A novelty detection approach has to motivate the reasons why a new operating condition is inferred.

Autoencoders (AEs) aim to learn some low-dimensional feature representation space on which the given data instances can be well reconstructed. This is a widely used technique for data compression or dimension reduction. AEs have been experimented with in the anomaly detection field. Here, the learned feature representations are enforced to learn the important regularities of the data to minimize reconstruction errors. Nevertheless, anomalies are difficult to reconstruct from the resulting representations and thus have large reconstruction errors.

In this paper, we propose to extend a generic AE model in order to make an effective implementation of the architecture template for finding novel operating conditions that satisfy all desiderata. An AE is composed of an encoding and a decoding network. The encoder maps the original data onto a low-dimensional feature space, while the decoder attempts to recover the data from the projected low-dimensional space. The parameters of these two networks are automatically learned, without any user intervention, through a reconstruction loss function. The main extensions we propose are shown in Figure 1 and consist of a Scorer that analyzes the reconstruction error (the error made by the decoder component in reconstructing the multivariate time series provided as an input); a Classifier that exploits the low-dimensional encoding of the input to recognize the actual operating condition; and a Localizer that recognizes the signal mainly supporting the inference of the novel condition, thus explaining the model behavior. A scaler is inserted at the beginning of the pipeline to normalize the input signals and evaluate their contribution without the biases that would result from different range values.

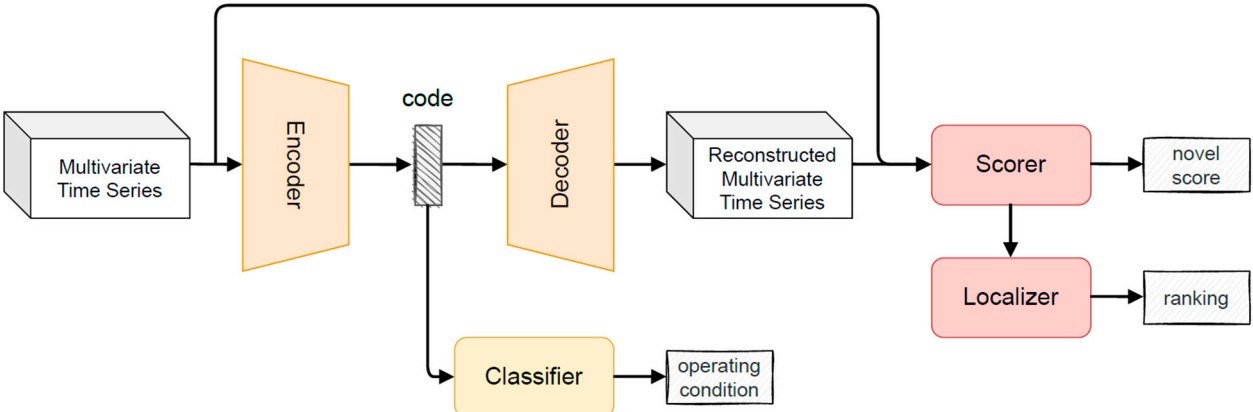

**Figure 1.** Autoencoder Architecture Template.

The Scorer is based on the computation for each signal of the mean absolute error (MAE) that measures the extent to which the generated output intensities are symmetrically close to the input intensities. In this way, a higher-than-training intensity is penalized by the same amount as an equally valued lower intensity. The evaluation is applied to all features and the average MAE is computed to be directly used as a novelty score. A novel condition is detected if the score is greater than a user-defined threshold. In our experiments, we adopted a threshold based on the maximum reconstruction error computed on the validation set. This value is increased by an $\alpha$ score (we experimented with two values of $\alpha$ score in our implementation, 0.02 in operating conditions with anomalies, 0.10 in the other scenarios) to reduce the number of false positives.

We provide two implementations for the Classifier, based on a supervised and an unsupervised approach, respectively. The supervised approach is based on a feed-forward neural network that we train in our implementation on the series generated by the encoder and on labels provided by experts to identify the operating conditions. The unsupervised approach is always built upon the series generated by the encoder and relies on a clustering algorithm. The optimal number of clusters (i.e., operating conditions) is automatically selected through the elbow method. In Section 4, we experiment with three metrics for the creation of the clusters.

The Localizer relies on the reconstruction error to compute the signals that mainly contributed to the inference. This component computes the error on the raw input signals, and not on the transformed version as for the Scorer. In this way, the impact of the normalization processes introduced by the data transformer component and the AE network is reduced and the error computation is more reliable. Figure 2a,b show the reconstruction of the input signals performed by the localizer. The plots refer to a CNN-based AE, trained on the signals from the operating conditions C1, C2, and C4 and aimed at identifying the

condition C3 as a new condition. Figure 2a refers to the reconstruction of the input signals of a time series of status C2 (composed of 9 input signals), which are part of the series used for the training. Figure 2b shows the original and reconstructed signals on C3. The number on the top of each plot is the mean reconstruction error. Figure 2a shows that the model can learn the input signals, and the error is generally low. Figure 2b shows that the reconstructed signals are not able to infer the actual signal. In particular, the higher error is measured on Y3, which represents one of the main reasons for the prediction of C3 as a novel state.

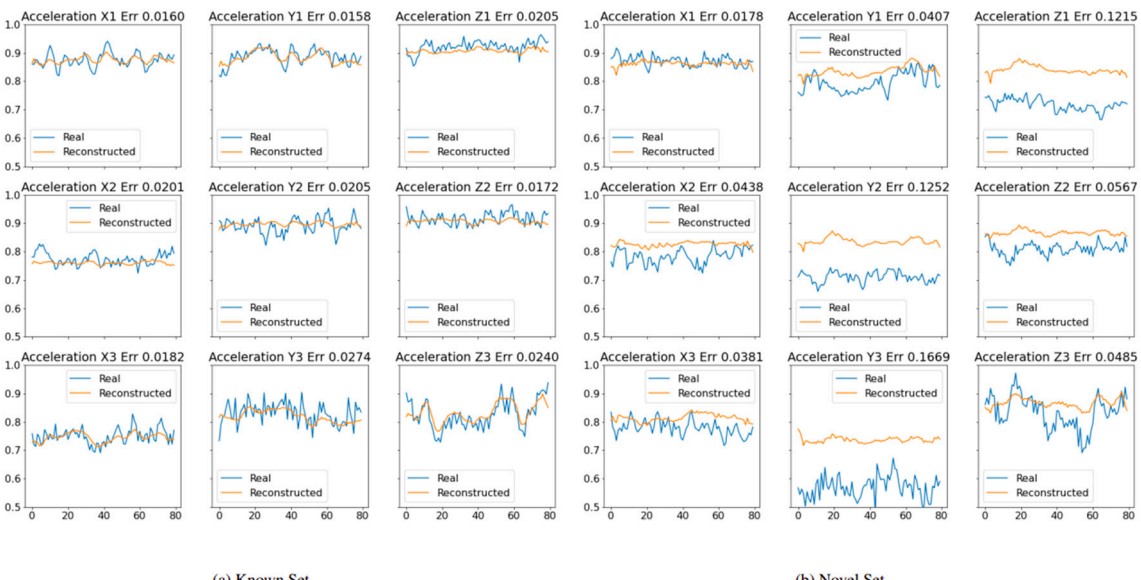

(a) Known Set

(b) Novel Set

**Figure 2.** The reconstruction performed by the Localizer of signals describing an already seen (**a**) or a novel operating condition (**b**).

## 4. Experimental Environment

To experiment with the architecture template described in Section 3, we implemented four AE versions, each one using a different DL technology. In particular, Section 4.1 introduces our implementations based on (1) a Feed-forward fully connected Neural Network; (2) a Convolution Neural Network; (3) a LSTM Neural Network; and a BiLSTM Neural Network. The experiments are performed against the dataset described in Section 4.2 obtained through an experimental platform developed in the Department of Industrial Engineering of the University of Bologna.

### 4.1. The AE Models

The experimentation of four models is justified by the trade-off they offer between complexity (in terms of the amount of data required and time/computational power for performing their train) and accuracy of the results. We expect that better results will be given by RNN AEs since they are based on networks specially designed for managing data sequences. Nevertheless, they are heavy to train on larger training sets and longer input sequences than other kinds of models. FC AEs are the simplest ones, but their use in many domains showed a correlation between the results achieved and the window size [23,24]. CNN AEs seem to offer an excellent compromise between simplicity and quality of results. Through the padding and kernel size, the input signal is split, thus creating an encoding that can potentially better generalize than FC approaches by identifying many patterns in the multivariate series. All architectures are trained for 300 epochs by using an Adam optimizer with a learning rate of 0.0004, a batch size of 64, and an early stopping strategy after 20 steps. Thus, in order to avoid the exploding gradients problem, we apply gradient clipping and normalization, which is quite common in long sequences.

Fully Connected AutoEncoder. A Fully Connected (FC) AutoEncoder [8] relies on Multilayer Perceptrons, or MLPs, to encode and decode the input data and intermediate representations respectively. During the encoding and decoding phase, the time series moves from the input layer through the hidden layers to the output layer, and the reconstructed signal of the output layer is measured against the original one. Then, it learns the time series representations by equally considering all time points without analyzing the signal temporality.

Experimented implementation. The FC-AE is composed of 3 layers for the encoder, with 200, 200, and 100 neurons; and 3 layers for the decoders with 100–200 neurons in the first two layers and 200 neurons per signal in the last layer. Note that the number of neurons in the last layer of the decoder depends on the selected window size. In our evaluations, we experimented the approach with several window sizes. Each layer applies a "tanh" activation function and a dropout of 0.2 is applied to the first layer for the encoding and decoding phases.

CNN AutoEncoder. A CNN AutoEncoder [10] applies temporal convolutive filters to an input organized in a grid to derive an intermediate representation that encodes the spatial proximity information of the original data (encoding phase) and adopts an inverse strategy to re-expand this intermediate knowledge. This allows the model to learn position and scale patterns and to extract spatial information along the time dimension. The distinguishing aspect of this approach is to learn time patterns by looking at the neighbor elements in the same and in other signals.

Experimented implementation. The implementation is based on 2 temporal convolutions in the encoding step and 3 transpose convolutions in the decoding step, with 32, 16, 16, 32, 9 filters respectively (since the multivariate time series analyzed are composed of 9 signals). Each layer applies a "relu" activation function and a "valid" padding and stride of 2 and a dropout of 0.2 is applied to the first layer in the encoding and the second layer in the decoding modules.

RNN AutoEncoder. The RNN AutoEncoder [9] (LSTM and BiLSTM in our case) is based on a recurrent connection of hidden representations generated from multiple MLPs, which is exploited to compress and reconstruct data while preserving their sequence and order of occurrence. RNNs show several nice properties, such as strong prediction performance as well as the ability to capture long-term temporal dependencies and variable-length observations. Recurrent neural networks explicitly handle the order in input observations. They learn long-term correlations in a sequence and are capable of accurately modeling complex multivariate sequences in several scenarios. However, the training time and the exploding gradient problem (i.e., when large error gradients accumulate and result in very large updates to neural network model weights during training) could become problematic for managing long time series.

Experimented implementation. We provide two implementations, one based on the LSTM technology, and the second on the BiLSTM. The LSTM-AE is based on stacked LSTMs with two layers in both encoding and decoding steps, with 200, 100, 100, and 200 neurons in each LSTM cell, and two fully connected layers at the end of the decoding phase with 16 neurons and 9 neurons (the number of signals in the dataset time series). Each layer applies a "tanh" activation function and a dropout of 0.2 is applied at the end of the encoding phase. The BiLSTM-AE is similar to LSTM-AE architecture, with the only difference being the application of a bidirectional connection in each stacked LSTM layer.

### 4.2. The Dataset

The platform used for the creation of the first dataset is composed of an asynchronous motor, a gearbox made of two pulleys that exchange the rotation through a belt, two shafts that share the motion thanks to a couple of gears, and an electromagnetic brake. The platform and its mechanical scheme are depicted in Figure 3. The platform is provided with three triaxial accelerometers, which are placed on the bearing's support, next to the second pulley and the two gearboxes, respectively. They have a sampling frequency of 12.8 kHz

per axis and an acceleration range of 500 Gpeak. A complete description of the platform can be found in [11]. For the purposes of experimentation, tests in four distinct operating conditions and a fault condition were conducted. The rotational speed is fixed at 660 rpm, while the distance between the pulley and the braking torque varies. The parameters and the duration of each condition are shown in Table 1. Note that each batch has a length of 10 min. A representation of the raw signals in the four operating conditions is provided in Figure 4. The considered signals represent a multi-variate series where each feature is an acceleration. As can be seen, while the accelerations in the first operating condition are rather stable, significant oscillations occur in the other conditions, but only state C4 describes an anomalous operation of the machinery (i.e., states C1–C3 represent normal operating conditions). The anomalies in state C4 are caused by the unforeseen overheating of the electric motor, which led to the sudden shutdown of the system.

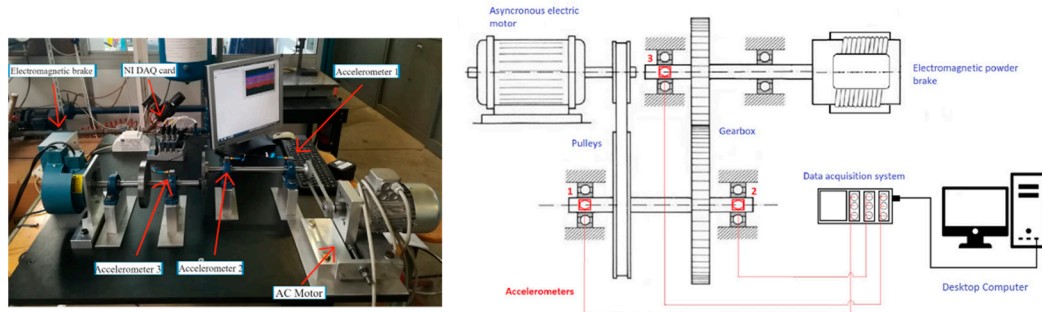

**Figure 3.** The test rig (**left**) and its mechanical scheme (**right**).

**Table 1.** Dataset Description.

| Operating Condition | Distance between Pulleys (mm) | Braking Torque (Nm) | Duration (min) |
|---|---|---|---|
| C1 | 27.33 | 0.1 | 70 |
| C2 | 27.33 | 0.5 | 150 |
| C3 | 27.54 | 0.1 | 70 |
| C4 | 27.54 | 0.1 | 30 |

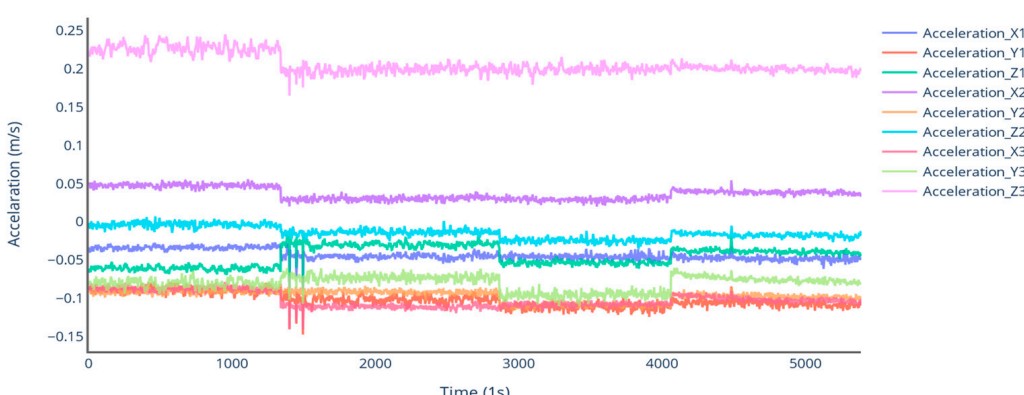

**Figure 4.** Raw signals from each operating condition (20 min per condition).

## 5. Experimental Evaluation

The experiments proposed in this section evaluated the ability of AEs in addressing the desiderata for a novelty detection system introduced in Section 3. In particular, there are three research questions that we address:

- (RQ1) Are AEs effective and robust in detecting novel operating conditions? (Section 5.1)
- (RQ2) Is it possible to classify the operating conditions of a machinery with an approach based on AEs? (Section 5.2)
- (RQ3) Can the AEs provide valuable explanations for their predictions? (Section 5.3)

## 5.1. Effectiveness and Robustness of the AEs in Novelty Detection

Three experiments are performed in this section to address RQ1. The first experiment (Section 5.1.1) evaluates the effectiveness of AEs in the novelty detection task. Then, two experiments evaluate the robustness of the approach in discriminating between anomalies and novel conditions (Section 5.1.2), and in working with varying window sizes (Section 5.1.3).

### 5.1.1. Evaluating the Novelty Detection Task in Normal Operating Conditions

Description. In this section, we evaluate only the first three operating conditions of the dataset, which refer to "normal" statuses of the machinery, where no anomaly is generated. We experimented with six scenarios (see Table 2), generated through all possible permutations of the conditions. In each scenario, the operating conditions described in the dataset are alternatively evaluated as part of the known set (i.e., the past operating condition) or the novel set. Scenario S12, for example, considers C3 as a known status, and C1 and C2 as novel conditions to detect. Two experiments were performed. In the first, the ability of the AE models is evaluated by detecting the novel state. We trained the AE models with the time series representing the known sets and we evaluated them with the ones in the novel sets. The training set constitutes 10 min of the known sets specified in the scenarios randomly selected. The test set is composed of the remaining data (from the known and novel sets). Column Novel in Table 3 shows the results of this experiment in terms of F1 score by considering a window size equal to 200 (100 s). In the second experiment, the ability of AEs to detect already existing operating conditions is evaluated. The input conditions belonging to the known sets are in this case also used to evaluate the approach. The training and test sets are built as for the previous experiment. Column Known in Table 3 shows the results of these experiments in terms of F1 score. Table 4 shows the F1 score obtained by averaging the results on the known and novel sets, thus providing an overall evaluation of the behavior of the approach.

**Table 2.** Scenarios used for the evaluation of the novelty detection task with no anomaly.

| Scenario | Known Set | Novel Set |
|----------|-----------|-----------|
| S12 | C3 | C1, C2 |
| S13 | C2 | C1, C3 |
| S23 | C1 | C2, C3 |
| S1 | C2, C3 | C1 |
| S2 | C1, C3 | C2 |
| S3 | C1, C2 | C3 |

**Table 3.** Evaluation of the effectiveness in normal operating conditions: F1 Score on the Known and Novel sets.

| Scenario | FC | | CNN | | LSTM | | BiLSTM | | PCA | | SVM | | Cluster | |
|----------|-------|-------|-------|-------|-------|-------|-------|-------|-------|-------|-------|-------|-------|-------|
| | Known | Novel | Known | Novel | Known | Novel | Known | Novel | Known | Novel | Known | Novel | Known | Novel |
| S12 | 99.22 | 99.83 | 99.29 | 99.84 | 99.34 | 99.85 | 99.60 | 99.91 | 5.90 | 90.21 | 15.86 | 0 | 66.57 | 94.69 |
| S13 | 100 | 100 | 100 | 100 | 100 | 100 | 99.98 | 99.98 | 8.36 | 65.56 | 32.82 | 40.80 | 75.07 | 82.02 |
| S23 | 100 | 100 | 100 | 100 | 100 | 100 | 100 | 100 | 0 | 87.57 | 80.07 | 95.49 | 91.90 | 97.92 |
| S1 | 100 | 100 | 100 | 100 | 100 | 100 | 99.81 | 99.47 | 69.77 | 61.21 | 48.76 | 0 | 99.64 | 99.02 |
| S2 | 100 | 100 | 98.74 | 99.08 | 100 | 100 | 99.89 | 99.92 | 86.31 | 92.03 | 39.68 | 22.47 | 100 | 100 |
| S3 | 100 | 100 | 100 | 100 | 100 | 100 | 100 | 100 | 68.95 | 55.09 | 62.24 | 51.47 | 100 | 100 |
| Average | 99.87 | 99.97 | 99.67 | 99.82 | 99.89 | 99.98 | 99.88 | 99.88 | 39.88 | 75.28 | 46.57 | 35.04 | 88.86 | 95.61 |

**Table 4.** Evaluation of the effectiveness in normal operating conditions: average F1 score.

| Scenario | FC | CNN | LSTM | BiLSTM | PCA | SVM | Cluster |
|---|---|---|---|---|---|---|---|
| S12 | 99.52 | 99.56 | 99.60 | 99.76 | 48.06 | 7.93 | 80.63 |
| S13 | 100 | 100 | 100 | 99.98 | 36.96 | 36.81 | 78.54 |
| S23 | 100 | 100 | 100 | 100 | 43.78 | 87.78 | 94.91 |
| S1 | 100 | 100 | 100 | 99.64 | 65.49 | 24.38 | 99.33 |
| S2 | 100 | 98.91 | 100 | 99.90 | 89.17 | 31.07 | 100 |
| S3 | 100 | 100 | 100 | 100 | 62.02 | 56.85 | 100 |
| Average | 99.92 | 99.75 | 99.93 | 99.88 | 57.58 | 40.80 | 92.24 |

The AE models are compared with three state-of-the-art approaches. Firstly, the Principal Component Analysis (PCA) is frequently used in exploratory data analysis because it reveals the inner structure of the data and explains the variance in the data. PCA looks for correlations among the variables and determines the combination of values that best captures differences in outcomes. For anomaly detection, each new input is analyzed, and the anomaly detection algorithm computes its projection on the eigenvectors, together with a normalized reconstruction error. The normalized error is used as the anomaly score. The higher the error, the more anomalous the instance is [25]. The One-Class Support Vector Machine is another unsupervised learning algorithm that is trained only on the "normal" data, in our case the negative examples. It learns the boundaries of these points and is, therefore, able to classify any points that lie outside the boundary as outliers [26]. Online clustering can be considered a distance-based novelty detection approach, in which the "normal" class is characterized by a small number of prototype points in the data space [9]. During the prediction step, the distance between the "normal" points and new points is computed. A threshold is fixed to determine whether the current pattern belongs to the same cluster as the normal one, or creates a new cluster.

Discussion. The average results reported in Table 4 show that our approach largely outperforms the state-of-the-art approaches in all scenarios. High F1 score values (100% in most of the cases) are measured with all deep learning implementations. The evaluation in Table 3 shows that, conversely from the competing approaches, the AEs perform well in both tasks of recognizing a novel or a known operating condition. We observe that, to be fair, we applied the same hyper-parameters to all AE approaches in all scenarios (i.e., the ones selected via a grid search analysis to maximize the overall accuracy on all scenarios). The performance of the AE approaches can be significantly improved by applying hyper-parameter fine-tuning to the specific datasets.

5.1.2. Evaluating the Novelty Detection Task with Anomalies

Description. This experiment extends the one in Section 5.1.1, by including C4 in the scenarios under evaluation. C4 describes a novel operating condition, but it differs from the other ones for the existence of anomalies in the signals. As in Section 5.1.1, we consider all possible scenarios created from all k-permutations of the time series describing the operating conditions (see Table 5).

Table 5 reports the results of the experiments for each scenario in terms of F1 score, by considering a window size equal to 200 (100 s) and the problem of detecting an existing operating condition (Known column) or a new operating condition (Novel column) separated. Table 6 averages the result obtained for each scenario.

Discussion. The introduction of an operating condition with anomalies decreases the overall quality of the results reached, as shown in Tables 6 and 7, even if in all scenarios they largely outperform the competing approaches. As expected, we observe the main decreases in scenarios where C4 is part of the training set (S123, S12, S13, S23, S1, S2, and S3). The FC AE is the model that mainly suffers the noise of the anomalies, showing a drop in the F1 score equal to 9.91% (from 99.92% to 90.01%). The other AE shows a very limited decrease (less than 5%). The baseline does not show the same behavior. Some of the approaches largely decreased the performance (the drop in the performance of the CLUSTER is more

than 33%). Only the SVM was demonstrated to be resilient to the anomalies, showing a small increment of the F1 score (from 40.8% to 41.37%). The breakdown of the results in Table 6 shows that for each scenario, almost all AEs outperform the baselines in both the detections of known and novel conditions.

**Table 5.** Scenarios used for the evaluation of the novelty detection task with anomalies.

| Scenario | Known Set | Novel Set |
|----------|-----------|-----------|
| S123 | C4 | C1, C2, C3 |
| S124 | C3 | C1, C2, C4 |
| S134 | C2 | C1, C3, C4 |
| S234 | C1 | C2, C3, C4 |
| S12 | C3, C4 | C1, C2 |
| S13 | C2, C4 | C1, C3 |
| S14 | C2, C3 | C1, C4 |
| S23 | C1, C4 | C2, C3 |
| S24 | C1, C3 | C2, C4 |
| S34 | C1, C2 | C3, C4 |
| S1 | C2, C3, C4 | C1 |
| S2 | C1, C3, C4 | C2 |
| S3 | C1, C2, C4 | C3 |
| S4 | C1, C2, C3 | C4 |

**Table 6.** Evaluation of the effectiveness in normal conditions with anomalies: F1 Score on the Known and Novel sets.

| Scenario | FC | | CNN | | LSTM | | BiLSTM | | PCA | | SVM | | Cluster | |
|----------|-------|-------|-------|-------|-------|-------|-------|-------|-------|-------|-------|-------|-------|-------|
| | Known | Novel | Known | Novel | Known | Novel | Known | Novel | Known | Novel | Known | Novel | Known | Novel |
| S123 | 35.19 | 94.14 | 100 | 100 | 28.62 | 91.90 | 96.41 | 99.89 | 23.94 | 89.45 | 0 | 41.19 | 5.68 | 0.70 |
| S124 | 100 | 100 | 99.79 | 99.96 | 99.88 | 99.97 | 100 | 100 | 5.90 | 90.88 | 14.94 | 0 | 66.57 | 95.07 |
| S134 | 99.09 | 99.10 | 94.72 | 94.34 | 99.38 | 99.39 | 99.66 | 99.67 | 8.36 | 68.48 | 32.55 | 46.44 | 75.07 | 83.89 |
| S234 | 100 | 100 | 100 | 100 | 100 | 100 | 100 | 100 | 0 | 88.44 | 80.07 | 95.83 | 91.90 | 98.08 |
| S12 | 92.08 | 98.06 | 98.88 | 99.70 | 97.44 | 99.32 | 97.92 | 99.46 | 52.50 | 75.36 | 1.44 | 0.99 | 37.28 | 21.38 |
| S13 | 100 | 100 | 100 | 100 | 100 | 100 | 100 | 100 | 100 | 100 | 47.33 | 43.56 | 70.42 | 4.21 |
| S14 | 96.78 | 92.24 | 95.46 | 88.52 | 98.43 | 96.42 | 98.83 | 97.38 | 69.77 | 66.53 | 48.03 | 12.40 | 99.64 | 99.22 |
| S23 | 47.25 | 43.33 | 99.27 | 99.76 | 96.90 | 98.95 | 96.57 | 98.84 | 69.28 | 83.23 | 92.02 | 97.66 | 39.32 | 0 |
| S24 | 99.95 | 99.97 | 97.97 | 98.65 | 99.53 | 99.69 | 99.73 | 99.82 | 86.31 | 92.82 | 39.49 | 32.16 | 100 | 100 |
| S34 | 96.38 | 89.07 | 95.68 | 86.56 | 96.02 | 87.8 | 95.36 | 85.94 | 68.95 | 61.65 | 62.24 | 58.15 | 98.82 | 96.77 |
| S1 | 100 | 100 | 100 | 100 | 100 | 100 | 100 | 100 | 92.53 | 83.46 | 58.34 | 0 | 86.28 | 17.07 |
| S2 | 99.00 | 99.21 | 85.72 | 85.20 | 97.01 | 97.56 | 97.11 | 97.76 | 91.88 | 92.72 | 38.81 | 18.85 | 60.75 | 0 |
| S3 | 100 | 99.99 | 100 | 100 | 100 | 100 | 99.84 | 99.42 | 87.49 | 0 | 63.64 | 51.13 | 87.76 | 0 |
| S4 | 95.51 | 43.96 | 97.84 | 75.23 | 98.14 | 79.83 | 94.19 | 56.83 | 89.88 | 33.79 | 60.69 | 20.31 | 96.68 | 9.75 |
| Average | 90.09 | 89.93 | 97.52 | 94.85 | 93.67 | 96.49 | 98.26 | 95.36 | 60.49 | 73.34 | 45.69 | 37.05 | 72.58 | 44.72 |

### 5.1.3. Window Size Invariance

Description. In this experiment, we evaluate the capability of the AE approaches of being invariant to the window size. Six window sizes are selected (ranging from 40 records per window–20 s to 360 records–180 s) and the same experiment for detecting the known and novel sets as in Sections 5.1.1 and 5.1.2 was performed.

Discussion. The results of the experiments show that AE-based approaches outperform the competing approaches independent of the window size. Nevertheless, they suffer from the noise that slightly decreases the average F1 score and increases the standard deviation. In detail, if we consider AEs and normal operating conditions (Table 8), the larger the window sizes, the better the results obtained. The same does not happen for the baseline, where the performance of SVM is window size invariant, and the other approaches decrease with the increase in the size. Moreover, the AE models offer low values for the standard deviation, thus showing that they do not depend on the data analyzed. The same does not happen for the baseline where the standard deviation assumes values greater than 25%.

**Table 7.** Evaluation of the effectiveness in operating conditions with anomalies: average F1 score.

| Scenario | FC | CNN | LSTM | BiLSTM | PCA | SVM | Cluster |
|---|---|---|---|---|---|---|---|
| S123 | 64.67 | 100 | 60.26 | 98.15 | 56.70 | 20.59 | 3.19 |
| S124 | 100 | 99.88 | 99.93 | 100 | 48.39 | 7.47 | 80.82 |
| S134 | 99.10 | 94.53 | 99.38 | 99.66 | 38.42 | 39.49 | 79.48 |
| S234 | 100 | 100 | 100 | 100 | 44.22 | 87.95 | 94.99 |
| S12 | 95.07 | 99.29 | 98.38 | 98.69 | 63.93 | 1.22 | 29.33 |
| S13 | 100 | 100 | 100 | 100 | 100 | 45.44 | 37.32 |
| S14 | 94.51 | 91.99 | 97.42 | 98.11 | 68.15 | 30.21 | 99.43 |
| S23 | 45.29 | 99.52 | 97.93 | 97.71 | 76.25 | 94.84 | 19.66 |
| S24 | 99.96 | 98.31 | 99.61 | 99.77 | 89.56 | 35.83 | 100 |
| S34 | 92.73 | 91.12 | 91.91 | 90.65 | 65.30 | 60.20 | 97.80 |
| S1 | 100 | 100 | 100 | 100 | 87.99 | 29.17 | 51.67 |
| S2 | 99.11 | 85.46 | 97.29 | 97.43 | 92.30 | 28.83 | 30.38 |
| S3 | 100 | 100 | 100 | 99.63 | 43.75 | 57.39 | 43.88 |
| S4 | 69.73 | 86.53 | 88.99 | 75.51 | 61.84 | 40.50 | 53.22 |
| Average | 90.01 | 96.19 | 95.08 | 96.81 | 66.91 | 41.37 | 58.66 |

**Table 8.** Average F1 score and standard deviation for each window size in normal operating conditions.

| Window | FC | | CNN | | LSTM | | BiLSTM | | PCA | | SVM | | Cluster | |
|---|---|---|---|---|---|---|---|---|---|---|---|---|---|---|
| | Avg | Std | Avg | Std | Avg | Std | Avg | Std | Avg | Std | Avg | Std | Avg | Std |
| 40 | 99.48 | 0.62 | 99.37 | 0.65 | 98.88 | 1.19 | 94.89 | 9.19 | 90.12 | 6.56 | 40.83 | 25.14 | 97.31 | 2.54 |
| 80 | 99.71 | 0.49 | 99.63 | 0.45 | 99.56 | 0.32 | 99.16 | 0.75 | 76.67 | 8.72 | 39.82 | 25.36 | 96.15 | 4.40 |
| 120 | 99.99 | 0.02 | 99.99 | 0.02 | 99.86 | 0.27 | 99.29 | 0.95 | 67.24 | 12.07 | 40.25 | 25.78 | 94.95 | 5.76 |
| 200 | 99.92 | 0.18 | 99.75 | 0.41 | 99.93 | 0.15 | 99.88 | 0.14 | 57.58 | 17.26 | 40.80 | 25.58 | 92.24 | 9.13 |
| 240 | 100 | 0 | 99.86 | 0.3 | 100 | 0 | 99.61 | 0.59 | 55.39 | 16.71 | 41.36 | 25.38 | 90.76 | 11 |
| 360 | 100 | 0 | 100 | 0 | 100 | 0 | 99.94 | 0.13 | 52.96 | 14.58 | 45.10 | 26.62 | 87.54 | 14.8 |
| Average | 99.85 | 0.22 | 99.77 | 0.31 | 99.71 | 0.32 | 98.80 | 1.96 | 66.66 | 12.65 | 41.36 | 25.64 | 93.16 | 7.94 |

A similar evaluation is possible for the analysis of the operating condition with anomalies (Table 9). We observe that the overall performances of the Aes decrease (with an increase in the standard deviation), the performance of SVM and PCA is stable in terms of F1 score, and shows an increase in the standard deviation for the PCA, and the CLUSTER approach largely decreases with a large increase in terms of standard deviation.

**Table 9.** Average F1 score and standard deviation for each window size in operating conditions with anomalies.

| Window | FC | | CNN | | LSTM | | BiLSTM | | PCA | | SVM | | Cluster | |
|---|---|---|---|---|---|---|---|---|---|---|---|---|---|---|
| | Avg | Std | Avg | Std | Avg | Std | Avg | Std | Avg | Std | Avg | Std | Avg | Std |
| 40 | 78.86 | 25.41 | 82.15 | 21.20 | 69.89 | 29.53 | 67.33 | 29.66 | 59.31 | 30.62 | 41.75 | 25.57 | 60.57 | 33.50 |
| 80 | 85.66 | 19.10 | 88.51 | 13.05 | 82.23 | 23.60 | 80.14 | 27.10 | 62.67 | 23.91 | 41.33 | 25.63 | 60.89 | 32.77 |
| 120 | 90.27 | 14.61 | 92.11 | 12.61 | 88.61 | 18.16 | 93.25 | 15.56 | 63.30 | 18.48 | 41.39 | 25.73 | 59.96 | 32.53 |
| 200 | 90.01 | 16.63 | 96.19 | 5.11 | 95.08 | 10.17 | 96.81 | 6.35 | 66.91 | 19.15 | 41.37 | 25.79 | 58.65 | 31.80 |
| 240 | 96.47 | 4.59 | 90.28 | 14.65 | 89.67 | 19.78 | 94.89 | 13.55 | 67.95 | 19.79 | 41.46 | 25.66 | 58.07 | 31.46 |
| 360 | 92.44 | 17.99 | 96.96 | 4.65 | 84.95 | 23.1 | 94.21 | 12.47 | 69.47 | 21.97 | 42.63 | 26.23 | 56.49 | 30.74 |
| Average | 88.95 | 16.39 | 91.03 | 11.88 | 85.07 | 20.72 | 87.77 | 17.45 | 64.94 | 22.32 | 41.66 | 25.77 | 59.11 | 32.13 |

### 5.1.4. Lesson Learned

The experiments demonstrate that AEs can effectively address novelty detection tasks, as they achieve the highest F1 scores in all scenarios outperforming the baselines. The experiments also demonstrate that AE is a robust technology: the results are not largely affected by the task addressed (know status or novel status), by the implementation

(four deep learning techniques are evaluated), the quality of the dataset (scenarios with and without anomalies have been tested), and the dimension of the signals (we evaluated the approach varying the window size).

From the industrial perspective, the proposed approach gives the opportunity to automatically assign a label to the acquired signals, indicating the correct machinery operating condition, regardless of the dataset available during the model training phase. In this way, the secrecy of the implemented machinery setting is kept, and only the change from one condition to another, both know and novel, is recorded in the dataset to facilitate the subsequent activities of fault detection and diagnosis. In addition, results show that AEs outperform baseline approaches in performing a simultaneous analysis at a system-level and component-level. Indeed, although it decreased with respect to the previous experiment, a higher F1 score is obtained, even if both nominal conditions corresponding to the system setting and the anomalous condition corresponding to the motor fault, are considered during model training.

## 5.2. Evaluating the Classification of the Operating Condition

In this section, the ability of AEs to detect the current operating condition is evaluated. Two experiments evaluate the unsupervised (Section 5.2.1) and supervised (Section 5.2.2) techniques.

### 5.2.1. Unsupervised Implementation

Description. This experiment aims to evaluate the implementations proposed for the unsupervised classifier. The model is based on a hierarchical clustering technique applied to the encoding of the time series generated by the AE encoder. The evaluation takes into account many settings by varying the AE model and the metrics adopted to evaluate the clusters. Tables 10–12 show the accuracy obtained by considering different window sizes, averaging the results obtained in all scenarios, and adopting the homogeneity score, rand score, and adjusted mutual information as metrics. The column RAW is the reference baseline obtained by applying the hierarchical clustering technique to the (original) input time series.

Discussion. The results show that the baseline is outperformed by our approach independently of the selected window size and metrics. The BiLSTM-based AEs provide the best results (by considering the average and the best performance obtained with the smallest window size) with all metrics. Note that, different to the AE methods, the RAW approach is almost not sensitive to the window size.

**Table 10.** Accuracy obtained by the unsupervised classifiers: Homogeneity Score.

| Window | Raw | FC | CNN | LSTM | BiLSTM |
|--------|-------|-------|-------|-------|--------|
| 40 | 83.77 | 89.14 | 85.35 | 82.04 | 100 |
| 80 | 83.67 | 93.68 | 86.66 | 94.87 | 94.34 |
| 120 | 84.47 | 98.5 | 88.44 | 88.64 | 96.49 |
| 200 | 86.35 | 97.43 | 86.02 | 84.07 | 94.71 |
| 240 | 83.93 | 93.27 | 88.03 | 95.90 | 93.86 |
| 360 | 83.57 | 98.74 | 89.74 | 92.68 | 97.54 |
| Avg | 84.29 | 95.13 | 87.37 | 89.70 | 96.16 |
| Std | 1.06 | 3.77 | 1.65 | 5.75 | 2.34 |

**Table 11.** Accuracy obtained by the unsupervised classifiers: Adjusted Rand.

| Window | Raw | FC | CNN | LSTM | BiLSTM |
|--------|-----|-----|-----|------|--------|
| 40 | 88.67 | 95.28 | 88.91 | 89.81 | 100 |
| 80 | 88.53 | 97.59 | 94.66 | 98.20 | 98.03 |
| 120 | 89.05 | 99.29 | 95.07 | 93.29 | 98.79 |
| 200 | 89.61 | 98.66 | 89.30 | 90.55 | 98.11 |
| 240 | 89.32 | 97.39 | 94.84 | 98.64 | 97.64 |
| 360 | 88.47 | 99.34 | 94.12 | 95.12 | 99.32 |
| Avg | 88.94 | 97.93 | 92.82 | 94.27 | 98.65 |
| Std | 0.46 | 1.54 | 2.89 | 3.74 | 0.89 |

**Table 12.** Accuracy obtained by the unsupervised classifiers: Adjusted Mutual Information.

| Window | Raw | FC | CNN | LSTM | BiLSTM |
|--------|-----|-----|-----|------|--------|
| 40 | 89.3 | 92.92 | 90.57 | 87.19 | 100 |
| 80 | 89.31 | 94.79 | 90.58 | 95.76 | 95.15 |
| 120 | 89.69 | 98.76 | 92.19 | 87.89 | 96.99 |
| 200 | 91.13 | 97.92 | 91.01 | 88.75 | 95.71 |
| 240 | 89.13 | 94.48 | 91.63 | 96.58 | 94.92 |
| 360 | 89.29 | 99.73 | 91.51 | 94.30 | 98.54 |
| Avg | 89.64 | 96.43 | 91.25 | 91.75 | 96.89 |
| Std | 0.75 | 2.73 | 0.64 | 4.26 | 2.03 |

### 5.2.2. Supervised Implementation

Description. This experiment evaluates a classifier trained on the encodings generated by the AE approach and using target classes provided by the user. In particular, the implemented classifier is based on a Feed-Forward Fully Connected Neural Network. The network is composed of 1 layer with 64 neurons. Table 13 shows the accuracy obtained as the mean on all scenarios with a window size equal to 200 (Acc 200), and the mean/standard deviation on all scenarios with all evaluated window sizes. Table 14 shows the F1 score in classifying the different operating conditions by considering a window size equal to 200 and all evaluated window sizes.

**Table 13.** Evaluation of the supervised classifier: Accuracy.

| Supervised | Raw | FC | CNN | LSTM | BiLSTM |
|------------|-----|-----|-----|------|--------|
| Acc 200 | 97.61 | 99.63 | 98.34 | 98.13 | 99.60 |
| Acc Avg | 97.48 | 99.08 | 98.78 | 98.52 | 98.86 |
| Acc Std | 0.24 | 0.58 | 0.66 | 0.26 | 0.51 |

**Table 14.** Evaluation of the supervised classifier: F1 score for each class.

| Conf. | Raw | FC | CNN | LSTM | BiLSTM |
|-------|-----|-----|-----|------|--------|
| C1 | 100 | 100 | 100 | 97.90 | 100 |
| C2 | 98.81 | 99.65 | 99.47 | 99.38 | 99.62 |
| C3 | 97.23 | 100 | 97.27 | 99.23 | 100 |
| C4 | 78.42 | 97.18 | 85.90 | 83.87 | 96.91 |
| Avg 200 | 93.62 | 99.21 | 95.66 | 95.10 | 99.13 |
| Std 200 | 10.19 | 1.36 | 6.61 | 7.51 | 1.49 |
| Avg all | 93.22 | 97.91 | 96.96 | 96.1 | 97.02 |
| Std all | 9.82 | 3.91 | 4.96 | 5.29 | 4.44 |

Discussion. The classifier shows a high accuracy score in all AE implementations. The low standard deviation measure shows the window size invariance of the approach.

As expected, the F1 score measured when detecting C4 is largely lower than the other operating conditions. This is due to the anomalies existing in the C4 operating condition that make the classification process more complex.

### 5.2.3. Lesson Learned

AEs can effectively recognize the operating conditions where the device is being worked on. The results registered in the experiments show high levels of accuracy and F1 score, making possible the application of the technology in real world scenarios. AEs largely overcome the competing approaches in both supervised and unsupervised scenarios. The approach is robust with respect to the window size selected.

From the industrial point of view, the AEs allow the detection of faults regardless of whether it has already occurred in the past. In particular, the supervised implementation is used for fault diagnosis, so that it is possible to stop the machinery before severe consequences occur once the failure is classified. On the other hand, when the unsupervised implementation is used and a novel condition is detected, an alarm can be generated to suggest a double check by the machinery operator. In the meantime, the observations are automatically labeled as "novel" to facilitate the subsequent model re-training.

### 5.3. Evaluation Localization Task

Description. The experiment evaluates the capability of the AE architecture to reconstruct the multivariate time series, using the reconstruction error for each feature of the time series, i.e., for each sensor of the machinery. This allows us not only to identify the presence of a never-seen-before operating condition, but also to identify the most anomalous signal that distinguishes the status. This evaluation allows us to understand if the experimented AE models can separate the features of the multivariate series and to identify the most anomalous signal with respect to the normality learned during training. The sensor that measures the strongest variation for each status has been identified in the laboratory. Table 15 provides a simple analysis of the signals for each operating condition. In particular, it shows the minimal distance from each signal in each operating condition to the other signals in the other conditions. The higher the distance, the easier the task of recognizing the most meaningful feature. The difference between the maximum and the minimum distance roughly shows how easy the task is to perform. According to the distance, the identification of the most relevant signal is easier for condition C1 (and then C2) than for the other conditions. The experiment consists of the selection of the signal that maximizes the reconstruction error. Table 10 shows the results in terms of precision at the first, third, and fifth levels.

**Table 15.** Input signals on the operating conditions.

| Features | C1 | C2 | C3 | C4 |
|---|---|---|---|---|
| Acceleration X1 | 0.009515 | 0.002341 | 0.002009 | 0.002009 |
| Acceleration Y1 | 0.012629 | 0.006512 | 0.003413 | 0.003413 |
| Acceleration Z1 | 0.005493 | 0.009277 | 0.005493 | 0.009277 |
| Acceleration X2 | 0.008993 | 0.002243 | 0.002243 | 0.006444 |
| Acceleration Y2 | 0.000181 | 0.000181 | 0.012083 | 0.005866 |
| Acceleration Z2 | 0.009027 | 0.005133 | 0.004702 | 0.004702 |
| Acceleration X3 | 0.015048 | 0.003005 | 0.003005 | 0.004736 |
| Acceleration Y3 | 0.003957 | 0.003795 | 0.015512 | 0.003795 |
| Acceleration Z3 | 0.024635 | 0.001038 | 0.001339 | 0.001038 |
| Avg | 0.009942 | 0.003725 | 0.005533 | 0.004587 |
| Std | 0.007089 | 0.002850 | 0.004936 | 0.002459 |
| Min | 0.000181 | 0.000181 | 0.001339 | 0.001038 |
| Max | 0.024635 | 0.009277 | 0.015512 | 0.009277 |
| Diff | 0.024455 | 0.009096 | 0.014174 | 0.008238 |

Discussion. Table 16 shows a large dependence of the results from the DL model used and that only in a few configurations (FC for condition C1, FC and LSTM for C3) is the approach able to identify as the first result the most relevant signal. A different situation occurs if we consider the first three results (p3): for all operating conditions, there is almost an implementation that obtains a high precision level (greater than 0.9). Moreover, as we were expecting, the performance on C4 and on C2 was lower than on the other conditions. In particular, on C4, only the CNN-based AE obtained a high precision score. This is due to the particular implementation of CNNs that base (through the padding and the kernel) their learning approach on all the signals simultaneously.

**Table 16.** Precision @k for most anomalous signal.

| | C1 | | | | C2 | | | | C3 | | | | C4 | | | |
|---|---|---|---|---|---|---|---|---|---|---|---|---|---|---|---|---|
| Precision | FC | CNN | LSTM | BiLSTM | FC | CNN | LSTM | BiLSTM | FC | CNN | LSTM | BiLSTM | FC | CNN | LSTM | BiLSTM |
| p1 | 93.84 | 1.89 | 87.88 | 33.78 | 54.99 | 67 | 39.77 | 34.31 | 100 | 40.38 | 95.02 | 12.77 | 0 | 59.57 | 5.91 | 0 |
| p3 | 99.94 | 20.33 | 100 | 100 | 91.77 | 94.48 | 87.38 | 82.47 | 100 | 100 | 100 | 36.27 | 0.35 | 94.36 | 17.11 | 4.89 |
| p5 | 100 | 58.85 | 100 | 100 | 98.74 | 99.28 | 98.96 | 96.95 | 100 | 100 | 100 | 72.62 | 25.62 | 99.74 | 26.06 | 18.87 |

Lesson Learned

The experiments show that AEs allow users to find the signals that have the largest impacts on the predictions, thus providing model explanations and allowing localization of the fault when it happens for the first time (novel condition).

## 6. Discussion

In this paper, a study on the methods used to determine the operating conditions in which machinery works is conducted. There are two relevant aspects to identifying a system's operating condition. First, it affects the behavior of components, meaning that their failure modes, progression, and thresholds depend on processed materials, products, or environmental factors. Although many efforts have been dedicated towards building models that are independent of the system's condition, industrial data collected from machinery are difficult to analyze if no information about the setting is provided. For instance, a sudden jump in the temperature values collected from an extruder, which in fact corresponds to a change in the processed material, might be confused with an anomaly if the information concerning the setting change is not provided. Second, the absence of information about the machinery setting may represent a limit for machinery producers that want to offer a full maintenance service to their clients. Machinery producers cannot know the actual operating conditions of their equipment, and their clients (machinery users) are not willing to share sensitive information that can be extracted from the knowledge of the implemented setting, e.g., the number of pieces per product type, or the production rate. Hence, datasets collected from machinery working in their operating environment may refer to different use modes, which are unknown to the producer.

This results in the impossibility of building general diagnostic and prognostic models that can be used for the same type of machinery operating in different environments. Therefore, it is fundamental to recognize when a setting change occurs in an unsupervised way or to determine whether the component behavior in that setting has already been analyzed (whether the operating condition is known). In this way, it is possible to automatically label the dataset with a class corresponding to the operating condition and increase knowledge about the machinery behavior in the actual operating condition.

The proposed autoencoder-based architecture allows the approach to perform three distinct tasks, i.e., classification, novelty detection, and localization, which support the collection of labeled datasets and the detection of novel situations. In particular, the classification task is to identify which operating condition among those known at the moment of the model training is implemented in any given moment. Novelty detection is to determine whether the operating condition implemented at that moment were not

included in the classification model training, and therefore is novel. Finally, localization is to determine which signals are most involved in the novelty detection.

As demonstrated in the previous section, the deep learning models outperform traditional methods in establishing the working condition of a given system. From experiments conducted in Section 5, it emerges that when no fault occurs in one of the components, the proposed approach classifies the distinct settings of the machinery with a high accuracy and precisely recognizes novel conditions with respect to those used during the model training (See Table 3). Similar results are obtained even considering a sudden fault in one of the components (See Table 5). In particular, if the fault (C4) is included in the training set, i.e., if the fault mode has already occurred in the past, both known and novel operating conditions are correctly detected, and the autoencoder-based approach provides better results than traditional approaches. However, in cases where the fault is not known at the moment of training, the ability to detect the condition C4 decreases as more operating conditions are considered in the training phase (look at the last scenario of Table 4). This result suggests that the proposed approach is able to determine the operating condition of the system, but has to be improved in order to perform a system-level analysis and component-level analysis simultaneously. Improvements in this direction will be the objective of further research. From the experiment conducted in Section 5.1.2, it emerges that the window size does not affect the performance of autoencoder-based approaches, which implies a lower dependency of the model from the parameters set by the user.

In Section 5.2, the performance of the classification of the operating conditions are evaluated. The results show that the classification accuracy is high for all conditions except for condition C4. This means that when only considering a system-level analysis, the proposed approach recognizes exactly the current operating condition. In other words, a dataset collected during the machinery functioning in its actual environment is correctly and automatically labeled with the operating condition it refers to.

Finally, the performance of the localization task is evaluated. The goal of this analysis is to determine whether there exists a signal, or a subset of signals, that mostly contributes to the novel condition detection. This evaluation is important for two reasons. First, selecting a smaller subset of input signals would reduce the quantity of data to transfer and analyze. Second, it helps to identify which signals are more relevant for the setting recognition problem, which is particularly important in making the results of deep learning approaches easier to interpret. However, considering the results summarized in Table 10, it emerges that the most relevant signals depend on both the model used and the condition implemented. Hence, the localization task is useful in identifying the signals that determine the change of the setting, but cannot be used as a feature selection method.

## 7. Conclusions

In this paper, we proposed the adoption of autoencoder-based architectures for addressing the novelty detection task. We introduced an architecture template that extends autoencoders, providing novelty detection, classification of the operating condition, and explanation of the inference. Four implementations, with autoencoders based on different kinds of deep learning models, were evaluated on a complex dataset, composed of novel conditions and including anomalies. The experiments demonstrated that our proposal outperforms state-of-the-art techniques in recognizing and classifying operating conditions and is able to find the most important signals in the series. Further assessments in other scenarios will be conducted in the future to verify the generality of our proposal.

**Author Contributions:** Conceptualization, F.D.B., F.C., M.P., A.B. and F.G.; Methodology, F.D.B., F.C., M.P., A.B. and F.G.; Software, F.D.B.; Validation, F.D.B., F.C. and F.G.; data curation, F.C.; writing, F.D.B., F.C. and F.G. All authors have read and agreed to the published version of the manuscript.

**Funding:** This research received no external funding.

**Institutional Review Board Statement:** Not applicable.

**Informed Consent Statement:** Not applicable.

**Data Availability Statement:** Not applicable.

**Conflicts of Interest:** The authors declare no conflict of interest.

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
