# Peer review of "Novelty Detection with Autoencoders for System Health Monitoring in Industrial Environments"

_applsci, doi:10.3390/app12104931_

Round 1
Reviewer 1 Report
The authors have presented Novelty Detection with Autoencoders for System Health Monitoring in Industrial Environments. Its an interesting work. The paper has been written well. The reviewer has the following comments for improving the paper further.
1) It is essential to include the pictorial representations of experimental facility/machines/sensors in the paper.
2) What is meant by c1, c2, c3, c4 conditions. Include the description.
3) Need to include some fundamental discussions on novelty detection concept before using it in the experiments.
4) Need more clarity on the results, tables, and discussions.
Reviewer 2 Report
The paper aims to support the development of predictive maintenance needed in industrial contexts to predict the occurrence of a failure to minimise unexpected downtimes and maximise the useful life of components.
The needs analysis to support the research is broadly explained concerning the shortcomings that many maintenance sector employees face and the various benefits the new strategy for maintenance management could provide. However, it is not entirely clear how the proposed autoencoder-based architecture will address all the various levels at which changes in maintenance management are dealt with.
In terms of innovation, it is positive that a study on methods for determining the operating condition in which a machinery works is conducted, as the paper enables to consider several points of view and interpretations. Applicants' claim for innovation is plausible, even if this statement is not highly specific. Particularly, the paper does not present enough evidence of how the authors' research outputs will be distinguished from other offerings as more details are omitted.
Round 2
Reviewer 2 Report
The paper could be published as it is.